# Identification, Virulence, and Molecular Characterization of a Recombinant Isolate of Grass Carp Reovirus Genotype I

**DOI:** 10.3390/v13050807

**Published:** 2021-04-30

**Authors:** Weiwei Zeng, Sven M. Bergmannc, Hanxu Dong, Ying Yang, Minglin Wu, Hong Liu, Yanfeng Chen, Hua Li

**Affiliations:** 1Guangdong Provincial Key Laboratory of Animal Molecular Design and Precise Breeding, School of Life Science and Engineering, Foshan University, Foshan 528231, China; 2112059069@stu.fosu.edu.cn (H.D.); yy41613@fosu.edu.cn (Y.Y.); chyfwf@hotmail.com (Y.C.); okhuali@fosu.edu.cn (H.L.); 2Institute of Infectology, Friedrich-Loeffler-Institut (FLI), Federal Research Institute for Animal Health, 17493 Greifswald, Germany; Sven.Bergmann@fli.de; 3Fisheries Research Institute, Anhui Academy of Agricultural Sciences, Hefei 230031, China; mlwu2020@163.com; 4Inspection and Quarantine Academy of Science, Shenzhen 518045, China; liuhong_szgacc@sina.com

**Keywords:** grass carp reovirus, identification, virulence, complete genomic, recombinant

## Abstract

The hemorrhagic disease of grass carp (HDGC) caused by grass carp reovirus (GCRV) still poses a great threat to the grass carp industry. Isolation and identification of the GCRV genotype I (GCRV-I) has been rarely reported in the past decade. In this study, a new GCRV was isolated from diseased fish with severe symptoms of enteritis and mild hemorrhages on the body surface. The isolate was further identified by cell culture, transmission electron, indirect immunofluorescence, and SDS-PAGE electrophoretic pattern analysis of genomic RNA. The results were consistent with the new isolate as a GCRV-I member and tentatively named GCRV-GZ1208. Both grass carp and rare minnow infected by the GCRV-GZ1208 have no obvious hemorrhagic symptoms, and the final mortality rate was ≤10%, indicating that it may be a low virulent isolate. GZ1208 possessed highest genomic homology to 873/GCHV (GCRV-I) and golden shiner reovirus (GSRV). Additionally, it was found a 90.7–98.3% nucleotide identity, a 96.4–100% amino acid identity, and <50% identity with GCRV-II and III genotypes. Interestingly, the sequences of some segments of GZ1208 were similar to GCRV-8733/GCHV, whereas the remaining segments were more closely related to GSRV, suggesting that a recombination event had occurred. Bootscan analysis of the complete genomic sequence confirmed this hypothesis, and recombination events between 873/GCHV and other GSRV-like viruses were also accompanied by gene mutations.

## 1. Introduction

Grass carp (*Ctenopharyngodon idella*) is economically the most important freshwater fish in China [1,2]. This fish species has been widely cultivated for the last 60 years and has been introduced into more than 40 countries [3,4,5]. It is now the most produced farmed fish species in the world [6]. However, grass carp farming is threatened by numerous viral and bacterial diseases [7,8,9], in particular hemorrhagic disease of grass carp (HDGC) caused by grass carp reovirus (GCRV). This is the most serious disease for grass carp that affects fingerling and yearling fish, resulting in enormous economic losses in the aquaculture industry [10,11,12,13,14]. HDGC was first described from Hubei Province in 1972. In 1984, the pathogen was determined to be GCRV [15]. The typical clinical symptoms of HDGC are hemorrhages in muscle, fins, gill covers, mouth cavity, and intestine tissues [10,12,16,17,18]. The degree of severity has been loosely classified into three types according to differences of clinical hemorrhagic symptoms as red muscles, red intestine, and red fin as well as red gill cover (operculum). However, these clinical symptoms are not pathognomonic, and they are especially not related to different GCRV genotypes [12,19].

GCRV was the first viral pathogen that was identified from aquatic animals in China in 1983 [13]. It is a member of the genus *Aquareovirus* in the family *Reoviridae* and a segmented ds RNA virus [20]. GCRV is the most virulent of the *Aquareovirus* members [21] that also can infect and cause disease in black carp (*Mylopharyngodon piceus*), the rare minnow (*Gobiocypris rarus*), and the top-mouth gudgeon (*Pseudorasbora parva**)* (Ding et al., 1991; Su et al., 2009; Jiang, 2009). In addition, GCRV can also infect silver carp (*Hypophthalmichehys molitrix*) and *Hemiculter leucisculus* without any clinical symptoms [22,23].

GCRV possessed a double-layered capsid, and its genome is composed of 11 double-stranded RNA segments with a total size of approximately 24 kbp. Currently, there are more than 40 isolates of GCRV from diseased but also healthy grass carp around the world [9,10,18,24,25,26]. Based on differences in genome constitution and sequences, GCRV has been classified into three genotypes: GCRV-I represented by GCRV-873, GCRV-II represented by HZ08, and GCRV-III containing only a single member of Hubei grass carp disease reovirus (HGDRV), formerly isolate 104 [10,18,24,27,28]. These three genotypes display great differences in their virulence level, cell culture characteristics, antigenicity, electrophoretic patterns of genomic RNA by SDS-PAGE, and genomic sequencing analysis [9,12,29,30]. These characteristics have been used to identify and classify viral isolates of GCRV.

Previous studies have demonstrated that most fish cell lines can be infected by genotype I, but only some HGDRV display an obvious cytopathic effects (CPE). Both genotypes are low or not virulent for grass carp [9,10,30,31,32,33]. In contrast, GCRV-II does not produce CPE, but most are highly virulent to grass carp with mortality levels exceeding 80% [4,9,18,26,30,34]. However, the GCRV-II isolate GD1108 and CQ1807 are non-virulent, although the reason for this effect is currently unclear [26,35].

Recent studies have demonstrated that isolates with different degrees of virulence were present in epidemics in China either as single or mixed genotype infections [9,10,24,31,33]. GCRV-II was associated with major hemorrhagic disease of grass carp (HDGC) outbreaks [18,36]. Occasionally, GCRV-I can be isolated from diseased grass carp [9,33], but both GCRV-I and GCRV-II have been implicated in recent cases of HDGC [24]. The GCRV-1 isolate GCRV-873 has been named grass carp hemorrhagic disease virus (GCHV) and was the first virus that was isolated and identified from an aquatic animal in China. GCRV-873 is also the most studied *Aquareovirus* member. However, the isolation, identification, and characterization of GCRV-I isolates have rarely been reported in the past decades, and only 14 isolates have completed genomes that are available for study in the GenBank (https://www.ncbi.nlm.nih.gov/genbank/; accessed on 16 November 2020). The availability of complete genomic sequences greatly benefits studies on morphological features and functional proteins as well as the geographical distribution and genetic evolution of aquatic viruses [37,38]. This lack of data is most likely a primary reason that the genetic evolution and pathogenesis of GCRV have not been fully elucidated.

The control of GCRV through vaccines has been met with some success for disease control in China. However, outbreaks continue sporadically and occur despite vaccination. The emergence of novel GCRV isolates with unclear epidemiology makes the complete prevention and treatment of GCRV problematic. Therefore, a large-scale epidemiological investigation can support the development and efficient usage of vaccines in specific infected areas. In this study, a novel GCRV was isolated from the moribund grass carp with severe hemorrhages in the intestine and only mild bleedings around the mouth and basal fins. There were no bacterial and parasitic infections detectable. We performed a thorough investigation of this isolate by cell culture, morphology, antigenicity, and genome analysis. Its virulence was evaluated in vivo additionally.

## 2. Materials and Methods

### 2.1. Virus Cell Line and Antibody

GCRV-873 (GCRV-I) was kindly provided by the Wuhan Institute of Virology, Chinese Academy of Sciences. HDGRV (GCRV-III) was a generous gift of the Yangtze River Fisheries Research Institute. GCRV-HZ08 (GCRV-II) was isolated and preserved by our laboratory. The Golden shiner reovirus (GSRV; ATCC VR-957) was purchased from the American Type Culture Collection (Manassas, VA, USA). The *Ctenopharyngodon idella* kidney (CIK) cell line derived from grass carp was used for the propagation of GCRV and GSRV [34]. Cells were maintained at 28 °C in M199 medium (Gibco, Grand Island, NY, USA) containing 10% fetal bovine serum (HyClone GE Healthcare, Logan, UT, USA) but used with 3% for virus propagation. The polyclonal antibody directed against GCRV-873 was prepared and preserved in the laboratory as previously described [30].

### 2.2. Virus Isolation

Virus isolation was performed as described with minor modifications [33]. In brief, diseased grass carp were sampled from a fish farm in Guangdong province. Liver, kidney, and spleen tissues were collected and homogenized in Dulbecco’s PBS, subjected to 3 freeze–thaw cycles and then centrifuged at 2880× *g* for 30 min at 4 °C. Then, the supernatant was filtered through a 0.22 mm membrane and stored at −70 °C. This served as the original virus isolate. CIK monolayers were grown in T-25 cell culture flasks (Corning Incorporated, Corning, NY, USA) and inoculated with 1 mL of 10-fold serial dilutions of the filtrated homogenate supernatant. The infected and non-infected cell cultures were incubated at 28 °C and examined daily for CPE over seven days. The viruses were passaged three times onto cell cultures. Control cells, infected and not infected, included those infected with GCRV-HZ08, 873, HDGRV, and GSRV (heterologous positive controls), while non-infected cells were used as a negative control.

### 2.3. Morphology of Virus Particles

GCRV morphology was examined using transmission electron microscopy (TEM) from cell culture suspensions where cells were scraped from the flask in terms of an obvious CPE observed or five days post infection (dpi). The cells were pelleted by centrifuge at 1.000× *g* for 5 min and embedded in epoxy resin as described previously [30]. Briefly, the pellets were fixed with 2.5% glutaraldehyde in 0.1 M sodium cacodylate pH 7.2 followed by 1% buffered osmium tetroxide (Polysciences, Warrington, PA, USA) and dehydrated with increasing concentrations of ethanol. The pellets were embedded in 100% Epon and left overnight. Thin sections (70 to 90 nm) were placed on Formvar-coated copper grids and stained with uranylacetate and then lead citrate. All micrographs were taken using an EM-100 CX II TEM (JEOL, Tokyo, Japan) operating at 80 kV.

### 2.4. Antigen Identification

Indirect immunofluorescence assays (IFA) were utilized to characterize viral isolates as described previously using a polyclonal antibody (see above) against GCRV-873 [39]. Briefly, CIK cells (2.5 × 10^5^ cells/mL) were incubated in 24-well plates (Corning Incorporated, Corning, NY, USA) and infected with suspensions of GCRV-873, GCRV-HZ08, HDGRV, and GSRV; then, they were fixed with ice-cold absolute methanol (−20 °C) for 10 min when obvious CPE was observed or five dpi followed by rinsing with PBS. The fixed cells were overlaid with PBS containing 5% BSA and incubated for 60 min at 37 °C for blocking. The polyclonal antibody (see above) was used at a 1:200 dilution in PBS. After 1 h incubation at 37 °C, the cell monolayers were rinsed three times with PBS, and the coverslips from each well were stained using 100 μL goat anti-rabbit IgG conjugated with fluorescein isothiocyanate (FITC) (Sigma Chemical, St. Lousi, MO, USA) and incubated for 1 h at 37 °C. Then, FITC-stained cells were rinsed with PBS, and 1000-fold dilution of propidium iodide was added to the cells and incubated for three min. Then, the preparation was examined immediately using a Nikon fluorescence microscope (Tokyo, Japan). Non-infected cells were used as control.

### 2.5. Virus Genome RNA Extraction and SDS-PAGE Analysis

Viral RNAs were extracted from infected cell culture supernatants by two freeze–thaw cycles followed by a centrifugation for 30 min at 2000× *g* to remove cellular debris. The supernatants were collected as a viral solution and centrifuged at 41,000 rpm using a Beckman Optima instrument followed by an SW41 rotor (Indianapolis, IN, USA) at 4 °C for 90 min. The pellet was resuspended in 1 mL TE buffer (0.01 M Tris, 0.001 M EDTA, pH 8.0). The purified virus was used for RNA extraction using Trizol^®^ Reagent (Invitrogen, Carlsbad, CA, USA) following the manufacturer’s recommendations. Viral genomic dsRNA were separated by SDS-PAGE on vertical slab gels (9% polyacrylamide) in Laemmli’s buffer and visualized by silver staining.

### 2.6. Regression Infection Test

*G. rarus*, the rare minnow, with average weights of 3.0 ± 0.5 g were kindly provided by the Institute of Hydrobiology of CAS, and *C. idella* (grass carp) with average weights of 25.0 ± 0.5 g were purchased from a fish farm located in Guangzhou. Fish were acclimatized at 28 °C under laboratory conditions for two weeks before the experiment started and then maintained in aerated water and fed daily with commercial dry feed pellets (Hello Fish Dry Pellets; CVM Products, Beijing, China). Possible viral contamination of fish and feed were checked by quantitative real-time PCR (qPCR) to verify that they were free from pathogens as previously described [40]. Care of animals was performed in compliance with the guidelines of the Animal Experiment Committee, South China Agricultural University (permit number: SYXK (Yue) 2014-0136). The new GCRV isolate GZ1208 was adjusted with M199 to a titer of 1 × 10^5^ TCID _50_/mL for its use in infection experiments. The rare minnow and grass carp were challenged by intraperitoneal injection with 0.05 mL and 0.2 mL diluted virus per fish, while controls were injected intraperitoneally with the corresponding volume of M199. There were 30 fish in each group, which were checked daily for clinical signs and mortality over 18 d. Tissue samples (pools of liver, kidney, and spleen tissues) of diseased and non-diseased fish for each group were collected for qPCR detection [40] and virus re-isolation onto CIK cells as outlined above at 18 dpi.

### 2.7. Full-Length Genome Amplification

Viral RNA was extracted from purified GCRV-GZ1208 using Trizol^®^ Reagent (see above) and full-length amplification of cDNA (FLAC) was used to amplify the entire genome of GZ1208 as described previously [41]. In brief, an anchor primer (5′-p-GACCTCTGAGGATTCTAAAC/iSp9/TCCAGTTTAGAATCC-OH-3′) possessed a C9 spacer between two complementary halves and a phosphorylated 5′ terminus. The anchor primer was ligated to the 3′ ends of the dsRNA segments using T4 RNA ligase (Takara, Tokyo, Japan). Then, the reaction mixtures were purified using a commercial RNA clean up kit (Qiagen, Hilden, Germany) and used as templates for first-strand cDNA synthesis using an AMV Reverse Transcription System (Takara, Dalian, China) and PCR amplification using a complementary primer (5′-GAGGGATCCAGTTTAGAATCCTCAGAGGTC-3′). PCR products were separated on a 1.2% agarose gel, and visible bands were purified using a Silica Bead DNA Gel Extraction kit (Fermentas, Waltham, MA, USA) and cloned into a pMD18-T vector (Takara, Dalian, China). Positive clones were sent to Sangon Biotech (Shanghai, China) for sequence analysis.

### 2.8. Sequence Analysis and Recombination Analysis

GSRV and other GCRV sequences used in this study for comparison were obtained from the GenBank database at the National Center for Biotechnology Information (NCBI). Nucleic acid and protein database searches were performed using BLAST at the National Centre for Biotechnology Information server. The DNA sequences and deduced amino acid sequences were analyzed using EditSeq (DNASTAR Lasergene 17.1.1. Madison, WI, USA). Multiple sequence alignments were performed using CLUSTAL W-2.1. Phylogenetic analysis was conducted via MEGA version 7.0.26 using the neighbor-joining method with 1000 duplicates, and bootstrap values >75% were considered statistically significant. Bootscanning analysis was performed using the Simplot 3.5.1 program with a 200 nt window moving in 20 nt steps.

## 3. Results

### 3.1. Virus Isolation

The homogenates obtained from the diseased fish were semi-purified and added to cultured CIK cells for purification and identification. Control cells showed no CPE and appeared healthy through these experiments (Figure 1A). CPE was observed as cell-to-cell fusions for the new virus isolate that produced large syncytia. Cells detached from the flask surface over the entire area and in the first round of virus propagation. It also appeared during the subsequent blind passages. These CPE were similar to those generated onto cells infected by GCRV 873 and by the other GSRV control, as indicated (Figure 1B–D). These characteristics were significantly different from those of HZ08 and HDGRV as well as the lack of CPE in HZ08-infected cells. The latter cultures possessed only a few pyknotic cells that gradually detached from the cell monolayer (Figure 1E). In contrast, large numbers of apoptotic cells aggregated and detached in cells infected with HGDRV (Figure 1F).

### 3.2. Morphology under EM

Numerous virus particles were observed from infected CIK cells generated by the addition of the tissue homogenate as well as control virus preparations. The virions of the new isolate were non-enveloped and nearly round-shaped with a diameter of 70–80 nm. This new virus formed viroplasms or inclusion bodies composed to paracrystalline arrays of virus particles that filled the entire cytoplasm (Figure 2A). This appearance was morphologically similar to GCRV-873 and GSRV (Figure 2B,C) but differed from HZ08 and HDGRV in that the viral particles were predominantly inclusion bodies in the cytoplasm (Figure 2D,E). Therefore, the new isolate was tentatively named GCRV-GZ1208.

### 3.3. Antigenic Identification of Isolate GCRV-GZ1208

We examined infected CIK cells for the reactivity to the polyclonal antibody, which was previously developed against the novel virus GCRV-GZ1208. Uninfected cells were completely absent of fluorescent signals using this antibody (Figure 3A). In contrast, strong fluorescence signals in cells infected with the tissue homogenate from clinical diseased fish were apparent (Figure 3B) as well as for GCRV-873 (Figure 3C) and GSRV (Figure 3D). Cells infected with GCRV-HZ08 (Figure 3E) and HDGRV (Figure 3F) lacked any fluorescent signal and were similar to the uninfected cells.

### 3.4. Genomic Electrophoretic Patterns

Genomic RNA of the new isolate GCRV-GZ1208 was compared with the genomes of GCRV-873, HZ08, HDGRV, and GSRV by SDS-PAGE analysis. The genomic RNA segments of GZ1208 were separated into 11 distinct bands similar to GCRV-873 and GSRV except for subtle differences. However, these were clearly distinct from HZ08 and HDGRV (Figure 4).

### 3.5. Virulence Analyses

We examined the new virus isolate using infections of healthy rare minnow and grass carp. At 18 dpi, the rare minnow and grass carp that had been injected with the GCRV-GZ1208 virus culture solution resulted in a cumulative mortality of 10% (3/30) and 3.3% (1/30), respectively. The dead fish had only mild hemorrhagic symptoms or no obvious clinical symptoms, but all dead fish were positive for GCRV-I like by qPCR, and a GCRV-I like virus was identified following re-isolation onto CIK cells. Furthermore, from 4/10 of rare minnow and 2/10 of grass carp, the genotype I GCRV-like virus was re-isolated. These results indicated that GCRV-GZ1208 was not very virulent.

### 3.6. Complete Genome Analysis

The complete genomic sequences of GCRV-GZ1208 genomic segments 1–11 (S1–11) were determined and deposited in the GenBank database under accession numbers KU240074–KU240084, respectively. The complete genome of this virus was 23,759 nt, and the 11 segments ranged in size from 820 to 3949 bp with a G+C content of 55.4% (Table 1). The genomic organization of GZ1208 appeared as the same as GCRV 873 and GSRV. One single open reading frame (ORF) was identified in each segment with the exception of S7 that was a functionally tricistronic gene that contained two ORFs that encoded NS4 and NS5, sequentially. The genome contained 12 ORFs encoding structural and nonstructural proteins typical of the genus *Aquareovirus*. The lengths of the GZ1208 non-coding regions (NCR) ranged from 12 to 42 bp at the 5′ ends and from 35 to 70 bp at the 3′ ends. All 11 GCRV–GZ1208 genome segments shared the 5′-GUUAUUU motif at the 5′-NCR and the 5′-UCAUC motif at the 3′-NCR and contained motifs that are very similar to the conserved ends of the GCRV 873 (or GCHV) and GSRV. We determined the nucleotide sequence length, ORF positions, predicted function of the encoded protein, and conserved terminal sequence of each segment (Table 1). A fusion-associated small transmembrane (FAST) protein was found in GZ1208, but a fiber-like protein was absent similar to GCRV 873 and GSRV. Alignment of the GZ1208 11 genome segments with other isolates of GCRV-I (873/GCHV), GCRV-II, GCRV-III (HGDRV), GSRV, and American grass carp reovirus (AGCRV) showed 90.7–98.3%, 33.6–53.8%, 35–50.6%, 91.5–99.2%, and 48.5–69.1% nucleotide sequence identities. The levels of amino sequence identity ranged from 14.6 to 100% (Table 1). In addition, a comparison of the entire genomes of GZ1208 and 873/GCHV (or GSRV) identified 2351 (or 1398) nucleotide differences and 72 (or 107) amino acid differences (Table 1).

### 3.7. Recombination Analysis

The genomic sequence of GCRV-GZ1208 was compared with that of other reference strains using the MAFFT alignment algorithm. Segment 5 of the GZ1208 isolate possessed the highest homology of 98.3% with GCRV-873/GCHV but only 91.7% with GSRV. Interestingly, some segments of the genome were the opposite, such as segment 9 of GZ1208 with 99.2% similarity with GSRV that was significantly higher than the 92.8% identity with GCRV-873/GCHV (Table 1). Genomic segments were similar indicating that the new isolate GZ1208 might have formed by recombination with similar isolates. Bootscanning analyses indicated the presence of six recombination breakpoints at nt 1069–1070, 1143–1144, 1228–1229, 1293–1294, 1503–1540, and 1645–1646 in segment 1; two recombination breakpoints at nt 1810–1811 and 1949–1950 in segment 5 and two recombination breakpoints at nt 350–351 and 370–371 in segment 7 were identified in the genome of the GZ1208 isolate (Figure 5). This analysis confirmed that the GZ1208 isolate was a novel recombinant isolate.

## 4. Discussion

Hemorrhagic disease of grass carp caused by GCRV remains the most serious infectious disease of grass carp and results in tremendous losses for the grass carp industry [14,26]. The three GCRV genotypes have proven to be highly genetically variable [10,18,24,27,28]. GCRV-I and II both cause hemorrhagic disease and are prevalent in grass carp [9,24]. However, most new isolates are GCRV-II and include GCRV-HZ08 [27,42], GCRV-GD108 [24], GCReV-109 [18], and AnH15 [25], while new GCRV-I isolates have rarely been reported. The identification of more new isolates assist studies to elucidate the diversity of GCRV and analysis of the molecular relationships among the genus [20]. In this study, a novel GCRV-I was isolated and identified, and its virulence was evaluated.

This novel virus generated obvious CPE using the supernatant of tissue homogenates obtained from diseased grass carp that were similar to GCRV-873 and GSRV but unlike those of HZ08 and HDGRV, implicating it as a GCRV-I member (Figure 1). The genomic morphology, genomic structure, and antigenicity were all consistent with this hypothesis. The presence of CPE has been linked to the presence of a FAST membrane protein [43], and we confirmed the presence of this feature in GZ1208 [43]. The morphology of the virions in the infected cells consisted of a mass of hexagonal non-enveloped virions with diameters of 70–80 nm, which was also consistent with the GCRV-1 group with minor differences and previous reports [30]. The presence of these minor differences for the particle arrays is unclear but is most likely related to intracellular replication. Additionally, the new virus possessed 11 dsRNA segments, which was more similar to 873 and GSRV but not exactly the same, and whole genome sequencing confirmed this. The polyclonal antibody against GCRV-873 also recognized GZ1208, 873, and GSRV but not GCRV-II isolates HuNan1307 or GCRV-III isolate HDGRV. This was further evidence that the new isolate GZ1208 was a GCRV-I member. These data were also consistent with the lack of cross-reactivity of polyclonal antisera prepared from all three genotypes, indicating the existence of at least three different GCRV serotypes as previously described [12].

GCRV is one of the most virulent agents of the genus *Aquareovirus* [44], and many GCRV have been reported to exhibit distinctive differences in virulence [10,18,24,26,33,44]. In general, genotype II is associated with high virulence and a short latent period, while genotype I is associated with low virulence and a longer latent period [45]. Grass carp is the natural host of GCRV, while the sensitivity of the rare minnow is higher. This species also can be used for GCRV as an important model animal to evaluate the GCRV virulence [23,46]. In our study, rare minnow and grass carp infected with GZ1208 did not exhibit hemorrhagic symptoms and resulted in a cumulative mortality of <10%. In addition, the detection and re-isolation of GZ1208 failed from the most survivors, which indicated either a lack of replication or immune clearance. In vitro infections of CIK cells indicated that GZ1208 was virulent. This may be an important feature of GCRV and GCRV-II isolates with low virulence to cells in vitro, but most of them are highly lethal to grass carp [18,24,30]. However, GCRV-I isolate infections to most fish cell lines can produce an obvious CPE with a strong virulence but has a low or no virulence to grass carp [9,10,30,31]. Earlier reports of GCRV-I isolates have indicated high virulence to grass carp with mortality levels 85% in fingerling and yearling populations [15,16,44,47,48], but recent reports have been the contrary [12,30].

Whole genome sequencing and analysis of GCRV-GZ1208 demonstrated that the genomic organization, size, and predicted functions of the 11 segments were similar to GCRV 873/GCHV and GSRV, and this included as well conserved terminal regions of viral dsRNAs in the *Reoviridae* family [20,29]. These conserved terminal sequences were identical for GCRV-873/GCHV, GZ1208, and GSRV, which is indicative for the same species. The genomic G + C content of GZ1208 was 55.4% and slightly higher than that of other members of the genus and may account for the slight differences in the genomic electrophoretic patterns [20]. The nucleotide sequence identities of GZ1208 and any GCRV-II and III isolates were all <55% identity (Table 1), although segments 10 and 11 of GZ1208 and AGCRV possessed >55% homology. All protein sequences were >95% homologous between GZ1208 and GCRV-873/GCHV or GSRV. Species demarcation criteria for the genus *Aquareovirus* and differences of <55% identity in segment 10 are members of the same species if they also possess >95% aa sequence identity in VP2 [49]. Based on these species demarcation criteria, GCRV-GZ1208, GCRV-873/GCHV, and GSRV belong to the same species. Interestingly, although the nucleotide sequence identity of many segments of GZ1208 and GCRV-873/GCHV was relatively low (S1 92.6%, S3 90.7%, and S9 92.8%), the amino acid identities were very high (S1 99%, S3 99.7%, and S9 100%), indicating that most of the changes were silent (Table 1). GZ1208 shares the highest homology with GCRV-873 and GSRV, but their genomes were still 2351 and 1398 nucleotide differences, respectively, suggesting that gene mutations are a source of the genetic diversity and evolution of the GCRV.

Recombination is a universal phenomenon for the *Reovirus* family [50,51]. A fundamental characteristic of reoviruses is that two distinct viruses can infect the same cell, and their genomes can recombine to generate novel viruses [52]. Recombination analysis of different genome segments among GCRV-873/GCHV, GSRV, and GCRV-GZ1208 indicated that S5 from GCRV-GZ1208 and GCRV-873/GCHV were more similar than those of GSRV. However, S7 and S9 of GCRV-GZ1208 and GSRV were more similar to each other than to GCRV-873/GCHV (Table 1). Recombination events may have occurred between GZ1208 and GCRV-873/GCHV and GSRV isolates, and bootscanning analyses confirmed the recombinant events existed in segments 1, 5, and 7 such that GZ1208 was most likely formed by recombination between 873/GCHV and/or other GSRV-like viruses. However, the numbers of complete sequences for GCRV-I types is limited, our analysis could result from this bias, and more complete genome sequences are required for a complete analysis.

## 5. Conclusions

In conclusion, a novel isolate GZ1208 of GCRV-I was successfully identified from diseased grass carp. The etiological and genomic characteristics of GZ1208 were similar to GCRV-873 and GSRV, and this new isolate demonstrated strong in vitro but weak in vivo virulence. The new isolate was also genetically diverse compared with the other GCRV isolates, and recombination and gene mutations may be the sources of the genetic diversity and the evolution of the GCRV and the GZ1208 isolate. This new isolate was most likely derived from recombination between 873/GCHV and other GSRV-like viruses as well as from gene mutations.

## Figures and Tables

**Figure 1 viruses-13-00807-f001:**
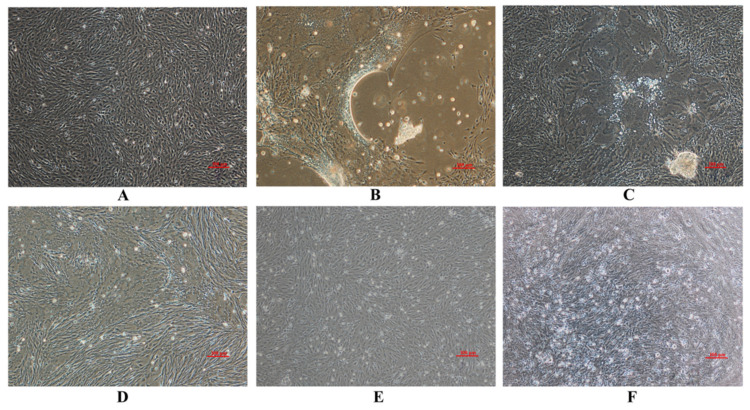
The CPE of C. idella kidney (CIK) cells infected with GCRV-GZ1208, 873, HZ08, HGDRV, and GSRV. (**A**) Uninfected CIK cells and cells infected with (**B**) GCRV-GZ1208 at 2 days post-infection (dpi), (**C**) GCRV-873 at 2 dpi, (**D**) GSRV at 2 dpi, (**E**) GCRV-HZ08 at 5 dpi, (**F**) HDGRV at 5 dpi.

**Figure 2 viruses-13-00807-f002:**
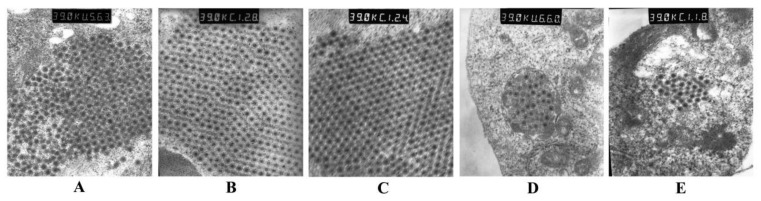
Ultra-thin sections from infected CIK cells: (**A**) GCRV GZ1208, (**B**) GCRV 873, (**C**) GSRV, (**D**) GCRV HZ08, (**E**) HDGRV × 39,000.

**Figure 3 viruses-13-00807-f003:**
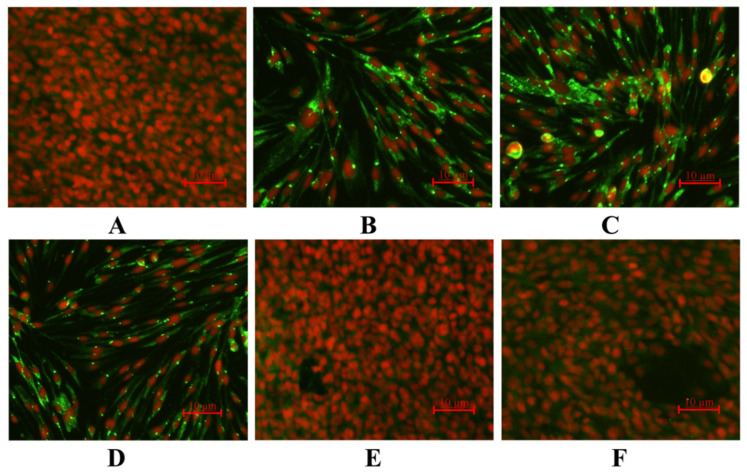
Indirect immunofluorescence assays for infected CIK cells. (**A**) Uninfected cells and cells infected with (**B**) GCRV-GZ1208, (**C**) GCRV-873, (**D**) GSRV, (**E**) GCRV-HZ08 (genotype II), and (**F**) HDGRV (genotype III).

**Figure 4 viruses-13-00807-f004:**
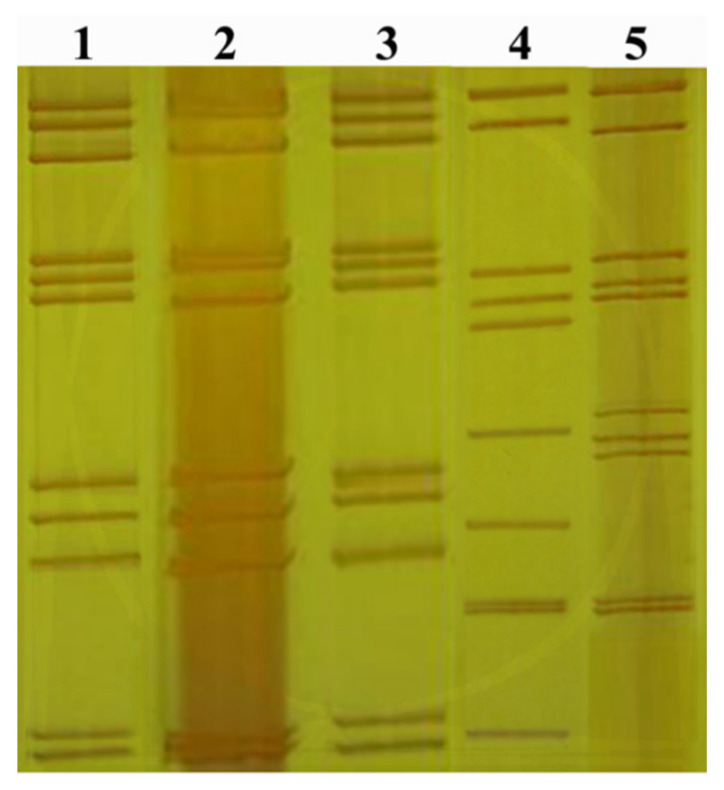
SDS-PAGE electropherotyping of GCRV and GSRV dsRNA genomic segments Lane 1 GCRV-GZ1208, Lane 2 GCRV-873, Lane 3 GSRV, Lane 4 GCRV-HZ08, and Lane 5 HDGRV.

**Figure 5 viruses-13-00807-f005:**
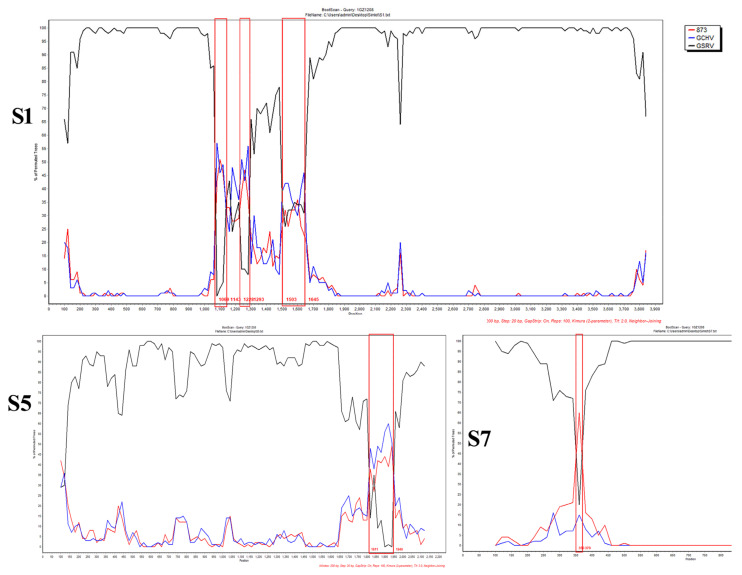
Recombination analysis by SimPlot of the genomic segments 1 (**S1**), 5 (**S5**), and 7 (**S7**) of the GZ1208 isolate. Six breakpoints in S1, two breakpoints in S5 and S7 of five recombination events were detected and highlighted in red boxes. The SimPlot was created using a window size of 200 bp and a step size of 20 bp.

**Table 1 viruses-13-00807-t001:** Characteristics of genome segments and percent identities of homologous genome segments and encoded proteins between isolate GZ1208 and other GCRV strains and GSRV.

Virus Strain	Segment	GenBank Accession No.	Length (bp)	Conserved Terminal Sequence	Predicted Function	Homology of Nucleotide (nu) and Amino Acid (aa) Sequence (%)									
						GCRV-I (873/GCHV)		GSRV		GCRV-II		GCRV-III (HDGRV)		AGCRV	
						nu	aa	nu	aa	nu	aa	nu	aa	nu	aa
GCRV-GZ1208	S1	KU240074	3949	GUUAUUU...UUCAUC	guanylyl transferase/methyl transferase VP1	92.6	99	92.4	98.7	44.6–44.9	31.2–31.5	43.2	32.1	61	63.9
	S2	KU240074	3877	GUUAUUU...UUCAUC	RNA-dependent RNA polymerase VP2	93.7	99.4	94.5	99.1	52.9–53.8	48.1–49.6	50.6	43.8	68.4	74.6
	S3	KU240074	3702	GUUAUUU...UUCAUC	NTPase/helicase VP3	90.7	99.7	92.7	99.5	45.2–46.3	34.5–35.7	48.2	35.6	69.1	76.8
	S4	KU240074	2320	GUUAUUU...UUCAUC	NS1	95.9	98	96.5	98.1	33.6–34.8	19.3–20.4	44.3 (S6)	19.4 (S6)	53.5	41
	S5	KU240074	2239	GUUAUUU...AUCAUC	putative core protein NTPase/VP5	98.3	98.8	91.7	97.1	38.8–39.9	31.5–32.6	46.4 (S4)	20.4 (S4)	58.9	57.5
	S6	KU240074	2103	GUUAUUU...UUCAUC	putative outer capsid VP4	94.2	99.7	93.4	99.2	45.3–46.2	31.6–32.9	41.6 (S5)	19.6 (S5)	66.9	70
	S7	KU240074	1414	GUUAUUU...UUCAUC	NS4	95.8	98.9	97.5	97.5	NE	NE	NE	NE	55.7	45.2
					NS5		98.5		98.1						
	S8	KU240074	1296	GUUAUUU...UUCAUC	core protein VP6	95.2	99	96.9	99.8	41.7–42.8(S9)	27.3–28.6(S9)	35	25.3	62.1	60.3
	S9	KU240074	1130	GUUAUUU...AUCAUC	NS2	92.8	100	99.2	100	42.1–43.4 (S10)	21.7–23.4 (S10)	NE	NE	60.4	57.8
	S10	KU240074	909	GUUAUUU...UUCAUC	outer capsid VP7	92.1	96.4	93.9	97.8	44.3–45.8 (S11)	14.6–16.2 (S11)	NE	NE	48.5	26.5
	S11	KU240074	820	GUUAUUU...UUCAUC	NS3	91.5	99.2	91.5	98	NE	NE	NE	NE	52.4	36.8

NE: no equivalent sequence. 873/GCHV: different names for the same.

## Data Availability

All of the materials and data that were used or generated in this study are described and available in the manuscript.

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
