# Peer review of "Identification, Virulence, and Molecular Characterization of a Recombinant Isolate of Grass Carp Reovirus Genotype I"

_viruses, 2021, doi:10.3390/v13050807_

Round 1
Reviewer 1 Report
The manuscript «Identification, virulence and molecular characterization of a recombinant isolate of grass carp reovirus genotype I” by Zeng et al describes the isolation and characterization of an isolate pf grass carp reovirus (GCRV) named GCRV‐GZ1208.
The manuscript needs a linguistic overhaul. It flourishes with spelling mistakes, sentences which are difficult to understand or don’t have sensible content, i.e. there are many sentences such as L 23 in abstract where you must guess the meaning. “Although neither species displayed obvious symptoms and was ≤ 10 % indicating a possible attenuated virus variant.” This makes it difficult to evaluate the manuscript.
Major objections
The authors use the expression “Recombination” and do not mention “reassortment”. I am confused about this. Do the authors mean that genomic segments of these reoviruses do not reassort between virus variants? Reassortment is a hallmark of virus families with segmented genomes like reoviruses. Or do you include reassortment in the expression recombination? This must be clarified. Recombination is either non-homogenous or homogeneous where parts of genomic segments is exchanged, which is a seldom phenomenon for reoviruses. However, the results presented indicate that you may have observed both recombination and reassortment.
There are very many abbreviations used for various isolates for GCRV-I, -II, -III; GSRV etc. For those readers not familiar to all GCRV variants and names there is a need to organise this in a table or similar where names of the isolates are listed according to which genogroup of GCRV they belong.
Fig 1. Here the cpe of various variants of GCRV are presented and compared. For the comparison to be meaningful the amount of infectious virus added, and dpi must be identical for the variants. This is not the case now.Minor objections
L 24 the abbreviation GSRV is not explained; L 62: HGDRV is not explained; L75 GCHD is not explained.
L35 L37 add “species” This fish “species”
L43 “ haemorrhages in” not “haemorrhaging of”
L49-50 use Italics of genus and virus family. There are other virus families that have dsRNA segmented genomes!
L51-54 and elsewhere: Put Latin names in parenthesis.
L118 stored at -170°C?
FIG 5. “Recombination breakpoints are indicated”? How- please explain
Reviewer 2 Report
As described in abstract, the genotype I of GCRV is rarely reported in the past decade, because it is not the pathogen caused HDGC at present which has been proven by many publications. So, is that possible the GZ1208 was isolated from the diseased fish with severe symptoms of enteritis? Regression infection test of the study also showed that, the isolate GZ1208 couldn't replicate similar symptoms. Considering the scientific meaning and the data consistent, I don’t think the manuscript is worthy to be published at the present status.
Round 2
Reviewer 2 Report
As described in abstract, the genotype I of GCRV is rarely reported in the past decade, because it is not the pathogen caused HDGC at present which has been proven by many publications. So, is that possible the GZ1208 was isolated from the diseased fish with severe symptoms of enteritis? Regression infection test of the study also showed that, the isolate GZ1208 couldn't replicate similar symptoms. Considering the scientific meaning and the data consistent, I don’t think the manuscript is worthy to be published at the present status.